# The Association between TRP Channels Expression and Clinicopathological Characteristics of Patients with Pancreatic Adenocarcinoma

**DOI:** 10.3390/ijms23169045

**Published:** 2022-08-12

**Authors:** Nicoleta-Raluca Chelaru, Andrei Chiosa, Andrei Sorop, Andreea Spiridon, Florentina Cojocaru, Dan Domocos, Dana Cucu, Irinel Popescu, Simona-Olimpia Dima

**Affiliations:** 1Center of Excellence in Translational Medicine, Fundeni Clinical Institute, 022328 Bucharest, Romania; 2Department of Anatomy, Animal Physiology and Biophysics, Faculty of Biology, University of Bucharest, Splaiul Independentei 91-95, 050095 Bucharest, Romania; 3Digestive Diseases and Liver Transplantation Center, Fundeni Clinical Institute, 022328 Bucharest, Romania

**Keywords:** TRP channels, PDAC, TRPV6, TRPM8, TRPA1, TCAF1, TCAF2

## Abstract

Pancreatic adenocarcinoma (PDAC) has low survival rates worldwide due to its tendency to be detected late and its characteristic desmoplastic reaction, which slows the use of targeted therapies. As such, the discovery of new connections between genes and the clinicopathological parameters contribute to the search for new biomarkers or targets for therapy. Transient receptor potential (TRP) channels are promising tools for cancer therapy or markers for PDAC. Therefore, in this study, we selected several genes encoding TRP proteins previously reported in cellular models, namely, Transient Receptor Potential Cation Channel Subfamily V Member 6 (TRPV6), Transient receptor potential ankyrin 1 (TRPA1), and Transient receptor potential cation channel subfamily M (melastatin) member 8 (TRPM8), as well as the TRPM8 Channel Associated Factor 1 (TCAF1) and TRPM8 Channel Associated Factor 2 (TCAF2) proteins, as regulatory factors. We analyzed the expression levels of tumors from patients enrolled in public datasets and confirmed the results with a validation cohort of PDAC patients enrolled in the Clinical Institute Fundeni, Romania. We found significantly higher expression levels of *TRPA1*, *TRPM8*, and *TCAF1/F2* in tumoral tissues compared to normal tissues, but lower expression levels of *TRPV6*, suggesting that TRP channels have either tumor-suppressive or oncogenic roles. The expression levels were correlated with the tumoral stages and are related to the genes involved in calcium homeostasis (*Calbindin 1* or *S100A4*) or to proteins participating in metastasis (*PTPN1*). We conclude that the selected TRP proteins provide new insights in the search for targets and biomarkers needed for therapeutic strategies for PDAC treatment.

## 1. Introduction

Pancreatic adenocarcinoma (PDAC) represents one of the most aggressive forms of cancer in adults and is the seventh leading cause of cancer-related death in both sexes [1]. PDAC survival has been recently reported as 41.7% at one year, 8.7% at three years, and 1.9% at five years, without a significant correlation between the disease stages and the overall survival rate [2]. Moreover, PDAC is curable only in a minority of patients due to its tendency to be detected late, as the majority are diagnosed with distant metastases.

Although alternative therapies are under development, the present treatment options for PDAC are limited to surgery followed by chemotherapy and radiotherapy or adjuvant therapy with FOLFIRINOX and gemcitabine regimens [3]. To date, modern targeted therapies, such as immunotherapy, have not been able to change patient outcomes [1]. Therefore, current research aims to meet the unmet need for novel therapeutic targets, as well as for biomarkers, to address all the pathophysiologic aspects of this devastating disease.

In the last decade, TRP ion channels were interrogated either as markers in tumoral cells or as putative targets, based on their draggability and functional expression in the membrane of tumoral cells.

In studies on PDAC, most of the analyses focused on the mechanistic roles of these channels in the migration, invasion, or growth of tumoral cells, but few have related TRPM8 protein expression to clinicopathological features. 

For instance, high *TRPM8* expression was significantly correlated with large tumor size, advanced TNM, and distant metastasis. Patients with high *TRPM8* expression had worse OS and DFS than those with low *TRPM8* expression, which confirmed *TRPM8* as a prognostic biomarker for OS [4]. Moreover, the TRPM8 protein was aberrantly expressed in the surgical specimens of pancreatic adenocarcinoma [5]. A recent bioinformatic analysis showed that *TRPM8* modulates immune infiltration and influences patient outcomes in many cancers, including PDAC [6]. Our published results point towards the functionality of TRPM8 channels and the importance of Ca^2+^ transports related to the migration and proliferation of PDAC cell lines [7].

A positive expression of the TRPV6 protein and mRNA level was significantly upregulated in pancreatic cancer tissues and was correlated with unfavorable patient survival rates [8].

In many cancers, including PDAC, TRPA1 stimulates tolerance to oxidative stress [9]. In our previous studies, we reported the expression of functional TRPA1 channels in PDAC cell lines [10].

In conclusion, all of these results consistently point towards some family members being important players in the tumorigenic process: *TRPM8*, *TRPM7*, *TRPV6*, and *TRPA1* [11]. Only recently has a study using the gene set enrichment analysis (GSEA) shown that two members of the TRPC subfamily (*TRPC3* and *TRPC7*) possess clinical predictive value for PDAC. Although insightful in providing new data about TRP’s contribution to pancreatic carcinogenesis, this report lacks mechanistic validation in cellular settings [12].

In a quest to complete our previous studies and previous reports regarding TRP channels in PDAC cell lines [7,8,10,13,14], in this study, we aimed to determine the expression levels of three TRP channels (*TRPM8*, *TRPA1*, and *TRPV6*) and associated factors *TCAF1* (*FAM115A*) and *TCAF2* (*FAM115C*) in the tumoral tissues of PDAC patients, and to correlate them with pathophysiological data. For this purpose, we used public databases and web-based tools for the analysis of RNA-sequencing data, as well as a validation cohort including PDAC patients from the Fundeni Clinical Institute. This analysis was intended to determine the best choices for a comprehensive evaluation of specific TRP channels and to probe a possible clinical translational application for these proteins in pancreatic cancer.

## 2. Results

### 2.1. Studies of the Selected TRP Genes in Patients Included in the TCGA and GTEx Public Datasets

To study the expression levels of TRP genes, we selected three family members previously analyzed in cell cultures that had consistent data regarding their functionality in tumorigenesis [5,8,10]. Thus, *TRPV6*, *TRPA1*, and *TRPM8* were the genes of choice, to which we added the genes encoding for the TRP channel-associated factors *TCAF1/FAM115A*, and *TCAF2/FAM115C*. To assess the expression of these genes, we used the TCGA, TARGET, and GTEx public datasets.

#### 2.1.1. Differential Expression Analysis of Selected TRP Channels 

First, using the GEPIA web platform as a bioinformatic tool, we explored the transcriptional levels of *TRPV6, TRPA1, TRPM8, TCAF1*, and *TCAF2* in pancreatic cancer (n = 179) and normal pancreatic tissues (n = 171). We found that *TRPA1, TRPM8, TCAF1*, and *TCAF2* expression levels were higher but nonsignificant in PDAC compared to normal tissues (*p* > 0.05). It is worth noting that all the genes possessed more transcripts in the pancreatic cancer tissues compared to normal tissues. Additionally, *TRPV6* expression was significantly down regulated in the PDAC samples compared to normal tissues (*p* < 0.05) (Figure 1A). We also noticed that TRPV6 expression was lower in tumoral tissues than in normal tissues. 

Next, we used UCSC Xena to explore the expression levels of *TRPV6, TRPA1, TCAF1, TCAF2*, and *TRPM8* in pancreatic cancer (n = 178) and in normal pancreatic tissue (n = 167). Consistent with the GEPIA results, the expression levels of *TRPA1, TRPM8, TCAF1*, and *TCAF2* were significantly higher in pancreatic cancer tissue (*p* < 0.001), and *TRPV6* expression was significantly lower in pancreatic cancer tissue compared to normal tissues (*p* < 0.001) (Figure 1B).

We also noticed that *TRPV6* expression was lower in tumoral tissues than in normal tissues when we applied the multiple gene comparison matrix plot from GEPIA. Moreover, out of all the TRP genes analyzed, *TRPV6* had the highest expression level (Figure 2).

#### 2.1.2. Selected TRP Channels Interact with Associated Factors and Calcium-Binding Proteins

We used GeneMANIA to create networks of the physical interactions and functions of the selected genes. As illustrated in Figure 3A, *TRPV6* interacts with 20 genes, among which the most important are *TCAF2*, *PTPN1* (Protein Tyrosine Phosphatase Non-receptor Type 1), and *CALB1* (Calbindin 1). *TRPA1* (Figure 3B) strongly interacts with *AKAP5* (A-kinase Anchoring Protein 5) and to a lesser degree with TRP ion channels from the TRPV subfamily. The gene encoding for the *TRPM8* channel is physically connected with *TCAF2* and interacts with *AKAP5* and *CHAMP1* (Figure 3C).

#### 2.1.3. Selected TRP Genes Differently Correlates with Tumor Stages and Metastasis

Furthermore, we evaluated the relationship between the TRP genes and the following clinicopathological features: tumor size, histologic differentiation, AJCC TNM staging, nodal status, and distant metastasis. For this analysis, we used the GEPIA and UALCAN web platforms. According to the AJCC (American Joint Committee on Cancer), the individual PDAC stages were divided into four stages (I–IV), and six sub-stages were classified as IA, IB, IIA, IIB, III, and IV.

First, we queried GEPIA to explore the relationship between the *TRPV6*, *TRPA1*, *TRPM8*, *TCAF1*, and *TCAF2* mRNA expressions and sub-stages in pancreatic cancer samples, compared to the control (normal subjects). We determined significantly lower expression levels for *TRPV6* mRNA in stages III and IV (Pr(>F) = 0.00708) (Figure 4A). Stage IV was also correlated with a significant increase of *TRPA1* (Pr(>F) = 0.0287 (Figure 4B) and of *TCAF1* (Pr(>F) = 0.00419; Figure 4D). Nevertheless, we found no significant differences in *TRPM8* mRNA expression and stages III and IV (Pr(>F) = 0.365) (Figure 4C) and *TCAF2* (Figure 4E).

We further analyzed the genes’ expression levels and pancreatic cancer sub-stages using the UALCAN platform, which employs a PERL *t*-test. The transcriptional level of *TRPV6* was statistically higher in stage II vs. stage III (*p* = 0.003) and higher in stage II vs. stage IV (*p* < 0.001) (Figure 4F). The transcriptional level of *TRPA1* was significantly associated with the stage of pancreatic cancer, with increased expression at a higher stage (Figure 4G). Among these, we found significant differences between stage II and stage III (*p* = 0.049) and between normal vs. stage III (*p* = 0.01). The analysis of *TRPM8* showed a statistically higher transcriptional level in stage II vs. stage III (*p* = 0.006) (Figure 4H). The expression level of *FAM115A* was found to be higher in stage I vs. stages II and III, but this was not statistically significant. Moreover, its expression was found to be higher in stage IV vs. all stages and normal (Figure 4I). Similarly, we observed that stage IV had the highest expression level of *FAM115C*, but this was not statistically significant (Figure 4J). 

We also examined the relationship between the genes’ expression levels and metastases using the TNMplot and Kruskal–Wallis test followed by the post hoc Dunn test. *TRPV6* gene expression was higher in normal tissues vs. in PDAC tissues (*p* = 2.25 × 10^−7^), in normal tissues vs. metastatic tissues (*p* = 3.20 × 10^−4^, in tumor tissues vs. metastatic tissues (*p* = 1.79 × 10^−8^), and in normal vs. metastatic tissues (Figure 4K). *TRPA1* gene expression was higher in normal tissues vs. tumoral tissues (*p* = 3.15 × 10^−1^), in tumor tissues vs. metastatic tissues (*p* = 2.02 × 10^−4^), and in normal vs. metastatic tissues (*p* = 2.70 × 10^−4^) (Figure 4L). 

The expression of *TCAF1* was significantly higher in tumoral vs. normal tissues (*p* = 4.61 × 10^−8^) and in tumoral vs. metastatic tissues (*p* = 1.78 × 10^−2^), but nonsignificant in normal vs. metastatic tissues (*p* = 3.94 × 10^−1^) (Figure 4N). *TRPM8* gene expression was significantly higher in tumoral vs. normal tissues (*p* = 2.06 × 10^−2^) and in normal vs. metastatic tissues (*p* = 1.61 × 10^−2^). However, we found a nonsignificant difference between its expression in tumor vs. metastatic tissues (*p* = 1.24 × 10^−1^), even though the differential expression was 0.46-fold change in metastatic versus tumor tissues (Figure 4M). To assess whether these results were reproducible, we further analyzed PDAC and non-tumoral adjacent tissues (NAT) in a cohort of Fundeni Clinical Institute patients.

### 2.2. Studies on the Validation Cohort of the Centre of Digestive Disease and Liver Transplantation in Fundeni Clinical Institute, Bucharest

To validate the results obtained from the public databases, we analyzed 218 tissue samples obtained after the surgery of 109 PDAC patients. Table 1 lists the clinicopathological characteristics of the patients included in this study. 

#### 2.2.1. Independent Validation of the Expression Level of the Selected Genes by qPCR

We used qRT-PCR to investigate the mRNA levels of *TRPA1* (n = 93), *TRPM8*, *TRPV6*, *TCAF1*, and *TCAF2* (n = 34) in paired surgical specimens from PDAC patients who had undergone aurgery at Fundeni Clinical Institute. The relative expression levels were significantly higher for *TRPA1* (*p* < 0.0001) with a median RQ value of 4.55, for *TRPM8* (*p* = 0.0064) with a median RQ value of 2.97, for *TCAF1* (*p* = 0.0028) with a median RQ value of 1.82, and for *TCAF2* (*p* = 0.0023) with a median RQ value of 2.81 in PDAC tissues compared to paired adjacent normal tissues. *TRPV6* expression was lower but nonsignificant in tumoral tissues compared to the paired adjacent normal tissues (*p* = 0.228) with a median RQ value of 0.78 (Figure 5).

#### 2.2.2. Clinicopathologic Characteristics of the PDAC Patients in the Validation Cohort

To evaluate the selected TRP gene expression levels, we analyzed 215 tissue samples obtained after surgery. Of 109 (16 in common among qRT-PCR and RNA-Seq) patients (56 males and 53 females), 73.40% had tumors with a size of ≥2 cm. The patients predominantly had early stage TNM, with 31.2% of the patients being stage I and 49.55% being stage II; the tumor differentiation was G1 in 40.74%, and the CA 19-9 values were above 36 U/mL in 77.9% (Table 1). To assess *TRPM8* and *TRPV6* expression, we examined 68 tissue samples representing paired NAT and tumoral tissues from 34 patients (26 males and 8 females). We added *TRP channel-associated factor 1/2* (*TCAF1/TCAF2*) genes to this analysis.

#### 2.2.3. Gene Expression Analysis of the TRP Family Members by RNA-Seq Confirms Overexpression of *TRPA1*, *TRPM8*, *TCAF1* and *TCAF2* and Lower Expression of TRPV6 in PDAC Tissues

To compare the results obtained from the analysis of the public databases, we used RNA-Seq data from 36 PDAC patients enrolled at the Fundeni Clinical Institute.

In the RNA-Seq analysis performed on the Fundeni Clinical Institute cohort, we found that *TRPA1* expression levels were significantly higher in PDAC tissues compared to NAT tissues (*p* < 0.0001). In the same line, the expression levels of *TRPM8* were significantly higher in PDAC tissues compared to NAT tissues (*p* = 0.0002). Moreover, we identified significantly increased levels of *TCAF1* and *TCAF2* in PDAC tissues (*p* = 0.0006 and *p* = 0.0079, respectively). Additionally, in our experiments, *TRPV6* expression levels did not significantly differ between tumoral tissues and NAT tissues (*p* = 0.999) (Figure 6).

#### 2.2.4. Correlation Analysis with Clinicopathological Characteristics of the PDAC Patients in the Validation Cohort

We found significant differences in the mRNA expression of *TRPV6* between the early PDAC stages (IA, IB, and IIA) with a median RQ value of 2.34 and stage IIB with a median RQ of 0.51 (*p* = 0.0017) (Figure 7A). Moreover, the patients with lymph node metastases (N1) had lower *TRPV6* expression with a median RQ of 2.34 compared to patients without lymph node metastases (N0) with a median RQ of 0.54 (*p* = 0.0004) (Figure 7B). 

*TCAF1* and *TCAF2* did not correlate with tumor stage or lymph node invasion (data not shown).

#### 2.2.5. TRP Protein Expression and Function in PDAC Tissues

The expression and functionality of the TRPM8 and TRPA1 channels in PDAC have been previously studied and reported by us and by others. In PDAC patients, the TRPM8 protein was overexpressed and correlated with gemcitabine resistance [15]. In our previous studies, we reported a higher expression of both TRPM8 (Figure 1, [7]) and TRPA1 (Figure 1, [10]) in PDAC cells than in the non-tumoral HPDE cell line. Using confocal microscopy, we noticed a clear expression of the TRPM8 protein at the membrane level (Figure 2, [7] and Appendix A) and a strong expression of TRPA1 at both the membrane and intracellular levels (Figure 5, [10] and Appendix A). The function of the channels was studied by challenging them with specific agonists, WS-12 for TRPM8 and AITC for TRPA1. These results were reproductible for this study, and the application of specific agonists determined a substantial Ca^2+^ entry, as shown by the imaging results (Appendix A). 

The TRPV6 protein and mRNA level were overexpressed in PC tissues compared to adjacent normal tissues (Figure 1, [8]) and were correlated with unfavorable survival in the PC patients. These results suggest that TRPV6 is an oncogene in PC and might participate in the development and progression of PDAC. 

To the best of our knowledge, the expression of TCAF1/F2 proteins in PDAC patients has not been reported in previous studies. To fill this gap, we analyzed the Human Protein Atlas (HPA) database. By inputting the target genes into the Human Protein Atlas database, the available representative immunohistochemistry images of TCAF1 and TCAF2 were outputted (Figure 8). 

## 3. Discussion

In this study, we report that the expression levels of the selected TRP channels (*TRPA1*, *TRPV6,* and *TRPM8*) and associated factors (the *TCAF1* and *TCAF2* genes) correlate with PDAC and the clinicopathologic characteristics of the disease. For a systematic and comprehensive analysis, we adapted the bioinformatics methods to evaluate the expression levels and possible prognosis of the selected genes in PDAC. We also compared the results obtained from public data with those of a patient cohort enrolled in Fundeni Clinical Institute. Below, we discuss the outcomes obtained for each of the TRP channels investigated.

### 3.1. Transient Receptor Potential Vanilloid 6 (TRPV6)

The analyzed TCGA and GTEx public datasets revealed that the *TRPV6* gene is abnormally lower in PDAC patients, suggesting a link between the downregulation of this gene and PDAC. We then examined the relationship between *TRPV6* and the clinical characteristics of PDAC patients and found that a lower expression of *TRPV6* is correlated with metastasis, advanced TNM stages of PDAC, and lymph node metastasis. The results obtained from the validation cohort confirmed the data analyzed from the public datasets. *TRPV6* expression predicted adverse prognosis after curative pancreatectomy and suggests that downregulation of the *TRPV6* may contribute to PDAC progression. In conclusion, we speculate that *TRPV6* may have a tumor-suppressive ability. This assumption contradicts a previous study performed by He Song et al. [5,8,10], who hypothesized the pro-oncogenic role of *TRPV6* based on overexpression in patients with pancreatic cancer and cell lines. Their study did not clearly distinguish the pancreatic cancer type, and the selected patients were not exclusively diagnosed with PDAC, contrary to the PDAC patients selected in our current analysis. Moreover, many of the patients were reported to be negative for *TRPV6* and were not included in their analysis [5,8,10]. However, most probably, *TRPV6* has pleiotropic functions, and its expression depends on the microenvironment, which is the underpinning factor in PDAC progression. 

The results of our study confirm the data reported by Zaccagnino A. et al. [16] from an in silico analysis of fresh frozen tissues obtained upon microdissection, which demonstrated that the expression of *TRPV6* was significantly lower in PDAC tissues compared to normal pancreatic tissue. Moreover, previous studies have reported the downregulation of the *TRPV6* gene in esophageal, non-small cell lung, and hepatocellular carcinoma, as well as in renal cancer [17,18,19]. 

*TRPV6* is also associated with other diseases, and its function is mainly related to calcium homeostasis. In our study, we performed a gene–gene network analysis to better understand the biological functions of the protein. As expected, we found that *TRPV6* is either co-expressed or physically related to calcium-binding proteins such as *calbindin 1* (*CAL1*) and *S100 calcium binding protein A10* (*S100A10*). However, the analysis revealed a strong interaction with *TCAF1*. In prostate cancer, TCAF1 negatively regulates cell migration when bound to TRPM8, but TCAFs are partner proteins for other ion channels, including TRPV6 [20]. The same type of regulation, decreasing TRPV6 membrane trafficking, is also possibly exerted in PDAC. 

Because the signaling pathways in which *TRPV6* participates are less described, we employed a GeneMANIA analysis to address this problem. We found that the protein *tyrosine phosphatase nonreceptor type 1* (*PTPN1*) has the strongest prediction for a common signaling pathway with TRPV6. In pancreatic cancer, PTP1B—the protein encoded by the *PTPN1* gene—was significantly overexpressed and was found to be associated with distant metastasis and tumor staging [21]. We are tempted to speculate that PTP1B and TRPV6 are connected through a pathway related to metastasis, but further investigations are needed to support this assumption.

We compared the results obtained from the analysis of public data and data from our cohort using RNASeq and qPCR. No statistical differences in *TRPV6* transcriptional levels were found between normal and PDAC tissue samples. We believe that these differences might reside in the type of tissue collected from the PDAC patients. While the public GTEx project has profiled samples from healthy individuals [22], we used adjacent non-tumoral tissues designated normal (NAT) for analysis. However, it has long been debated whether NAT is an intermediate, pre-neoplastic state and whether gene expression is enriched from stromal pathways [23]. Moreover, it is important to consider whether the tissue samples were collected from invasive or non-invasive parts of the tumor because *TRPV6* has preponderant expression in the invasive areas of breast cancer [24].

### 3.2. Transient Receptor Potential Ankyrin 1 (TRPA1)

TRPA1 has been shown to elicit an anti-migratory effect in PDAC, modulating various cellular processes such as cell cycle progression through non-selective calcium trafficking [10]. TRPA1 is the only member of the TRPA subfamily that is structurally specific for the large number of cytoplasmic ankyrin repeats, which make the channel prone to intracellular signaling, in addition to its transport function. 

In different cancer types, TRPA1 was found to be associated with oxidative stress tolerance [9] and inflammation [25]. In this study, we found that *TRPA1* expression was higher in pancreatic cancer tissues, which correlated with our previous results on PDAC cell lines. However, we reported in our previous study that TRPA1 decreases migration of Panc-1 cells, indicating these cells’ protective role, at least against migration [10]. This might explain the low expression of *TRPA1* in metastatic cells observed in this study. It is then possible that elevated *TRPA1* expression might restrain the metastasis of PDAC.

From the GeneMANIA analysis, we extracted the best correlation of *TRPA1* expression with the gene encoding *A-kinase anchor protein 5* (*AKP5*), which is a scaffolding protein of the cytoskeleton and has increased levels in stomach adenocarcinoma [26]. To date, we have no information about the role of AKP5 in PDAC or its relationship with TRPA1. The *TRPA1* gene is strongly co-expressed with members of the TRPV subfamily, sustaining the link between these two TRP subfamilies (TRPA and TRPV) in relation to pain and inflammation [27].

### 3.3. Transient Receptor Potential Melastatin 8 (TRPM8)

TRPM8 is the most studied TRP channel, because, out of all the family members, it has the highest levels of expression in different cancer types such as prostate cancer, colon cancer, breast cancer, melanoma, and PDAC [28,29,30,31]. The functions of TRPM8 are described mainly in relation to calcium homeostasis; as a second messenger, it regulates migration, proliferation, and apoptosis. In PDAC, its function was found to be associated with migration, proliferation, senescence, or tumoral stage [5,7]. Some studies have discovered posttranslational modification of TRPM8 and reported correlations of the channel function consecutive to glycosylation [13] or phosphorylation [32]. 

In this study, we found a direct correlation with the tumoral stage, with *TRPM8* being overexpressed in tumoral tissues compared to normal tissues, as reported in our previous studies on PDAC cell lines and in studies from other groups. The results obtained in GeneMANIA give further credence to earlier evidence that TRPM8 directly interacts with TCAF2, but, surprisingly, our data revealed no interaction with TCAF1. TCAF2 promotes cell migration in prostate cancer by recruiting TRPM8 to the cell membrane, whereas TCAF1 reduces the speed and migration of these cells [20]. It is thus reasonable to believe that high levels of *TRPM8* and *TCAF2* co-expressions in PDAC correspond to cancer invasiveness and metastasis. We also included the images provided by Human Protein Atlas in this study. The results showed that TCAF1 and TCAF2 are expressed in cancer tissues and are located at the cytoplasmic and membrane levels.

### 3.4. Channels Functionality

We explored in our previous studies, and this one, the function of TRPA1 and TRPM8 proteins. In cell cultures, the channels responded to specific agonists and showed substantial expression in a subpopulation of cells. It is noteworthy that not all cells expressed these proteins, as not all patients had significant increases of mRNA expression. Therefore, both TRPM8 and TRPA1 are linked to a specific condition at the cellular level, which may be related to migratory capacity or invasiveness. The function of these proteins that transport cations, and especially calcium, into the cells was explained in previous studies as being important in apoptosis. They reduce the expression of TRPV6-initiated apoptosis and cell cycle arrest [33]. Impaired regulation of intracellular calcium resulting from the overexpression of TRPM8 can cause cell proliferation and metastasis but can also enhance drug-induced cell apoptosis [34]. 

## 4. Materials and Methods

### 4.1. Differential Expression Analysis by GEPIA

The GEPIA (Gene Expression Profiling Interactive Analysis) platform is an interactive bioinformatics platform developed using the inclusion of high-throughput RNA sequencing expression data from TCGA (The Cancer Genome Atlas (https://portal.gdc.cancer.gov/ accessed on 20 May 2022)) and the GTEx (Genotype-Tissue Expression (https://www.gtexportal.org/ accessed on 20 May 2022)) projects [35]. The results shown here are based upon data generated by the TCGA Research Network: https://www.cancer.gov/tcga accessed on 20 May 2022. The GTEx Project was supported by the Common Fund of the Office of the Director of the National Institutes of Health, and by NCI, NHGRI, NHLBI, NIDA, NIMH, and NINDS. The data used for the analyses described in this manuscript were obtained on 20/DD/YY and/or dbGaP accession number phs000424.vN.pN on 20 May 2022. GEPIA is available at http://gepia.cancerpku.cn/index.html, accessed on 20 May 2022. In the present study, this online tool was used to analyze the expression profiles of *TRPA1*, *TRPV6*, *TCAF1*, and *TCAF2* in PDAC samples and normal pancreatic tissue samples based on the selected datasets of 179 PDAC patients (TCGA cohort). The multiple-gene comparison matrix plot was also applied to compare the expression levels of the studied genes. The transcripts per million (TPM) unit was used to measure RNA expression.

### 4.2. Differential Expression by UCSC Xena 

To further confirm the expression profile of the selected TRP- and TRP channel-associated factor gene panels, we corroborated our relative expression data with the UCSC Xena database (http://xena.ucsc.edu/, accessed on 22 May 2022), an online exploration tool for the visualization and analysis of both large public repositories and private datasets created by bioinformatics researchers at the University of California, Santa Cruz [36]. 

We selected the TCGA, TARGET, and GTEx studies from the Xena browser and applied genomic and phenotypic filters. Out of 350 samples, we excluded 1 metastatic sample and 4 solid-tissue normal samples, which were previously included in the GEPIA analysis. The TCGA, TARGET, and GTEx datasets were downloaded, and then we compared the previously mentioned gene expression panels of 178 PDAC tumor samples and 167 normal pancreatic tissue samples, using 345 samples in total. The transcripts per million (TPM) unit was used to measure RNA expression [37].

### 4.3. Differential Expression by Ualcan

Ualcan is a web platform (available at http://ualcan.path.uab.edu/, accessed on 15 May 2022) that allowed us to perform a gene expression analysis on the TCGA cohort, which we used to corroborate our data with previous results [38].

The dataset used comprised 178 PDAC samples and 4 normal samples resulting from the TCGA RNA-seq data.

### 4.4. Differential Expression by TNMplot

TNMplot (available at https://tnmplot.com/analysis/, accessed on 19 May 2022), is a web tool containing almost 57.000 samples from GTEx, TCGA, and TARGET RNA-Seq data, as well as from GeneChip, which provides gene expression comparisons from normal, tumor and metastatic tissues [39]. We queried the GeneChip dataset, which has over 3691 normal samples, 29,376 tumor samples, and 453 metastatic samples.

### 4.5. Interaction Network by GeneMANIA

The GeneMANIA Cytoscape platform (http://www.genemania.org accessed on 2 May 2022), is a web tool used for defining gene–gene or protein–protein interactions, providing information regarding the relationship of the queried subject or multiple queried subjects to the GeneMania interaction network [40].

### 4.6. Patients and Sample Preparation

In our study, we included patients surgically treated for PDAC between 2003 and 2019 at the Centre of Digestive Disease and Liver Transplantation in Fundeni Clinical Institute, Bucharest, Romania. Clinicopathological and follow-up data were gathered retrospectively for each case from medical records. Post-operative survival was measured from the day of surgery to the time of death or to a final follow-up visit on 14 August 2021. Prior to surgery, all the patients provided informed consent, which was approved by the local ethics committee of Fundeni Clinical Institute (23878/14 June 2018). For each patient, tumoral tissue (T) and normal tissue adjacent to the tumor (NAT), which represents an intermediate state between healthy and tumoral tissues, was obtained after surgery and collected in an RNAlater solution (Ambion, Applied Biosystems), then cryopreserved using the snap-frozen method in liquid nitrogen and further stored at −80 °C. 

### 4.7. RNA Sequencing Analysis (RNA-Seq)

The RNASeq analysis included 32 non-tumoral and 36 tumoral tissue specimens from 36 PDAC patients. The library was constructed using the Illumina TruSeq Stranded mRNA kit (Illumina) according to the standard manufacturer’s protocol (part no. 15031047 Rev. E October 2013). All the libraries were subsequently submitted to an Illumina NextSeq500 platform (Illumina), and sequencing was performed. The raw data were collected using NextSeq System Suite 2.2.0.4 and analyzed using Illumina BaseSpace Sequence Hub.

### 4.8. Quantitative Real-Time PCR Analysis

From each tissue sample collected post-operation (tumoral and adjacent non-tumoral tissue), we isolated total RNA with TRIzol™ Reagent (Invitrogen, Waltham, MA, USA) according to the manufacturer’s instructions. We used the High-Capacity cDNA Reverse Transcription Kit (Applied Biosystems, Waltham, MA, USA) to perform RT-PCR to obtain cDNA. qPCR amplification was performed using the TaqMan Universal PCR Master Mix (Applied Biosystems, Waltham, MA, USA), and each sample of tumoral and adjacent non-tumoral tissue was analyzed in duplicate. The levels of each TRP gene, namely, *TRPM8* mRNA (ID: Hs01066596_m1, Thermo Fisher Scientific) which was pre-amplified with the TaqMan™ PreAmp Master Mix Kit (cat no. 4384267); *TRPA1* mRNA (ref Hs00175798_m1, Thermo Fisher Scientific); *TCAF1* mRNA (ref. Hs00206993_m1, Thermo Fisher Scientific); *TCAF2* mRNA (ref Hs04186295_m1, Thermo Fisher Scientific); and *TRPV6* mRNA (ref Hs00367960, Thermo Fisher Scientific) were normalized to housekeeping gene *18S* (ID: Hs99999901_s1, Thermo Fisher Scientific). The results obtained using the SDS 1.4 software were calculated with the comparative Ct method 2^−ΔΔCt^ and 2^−ΔCt^.

### 4.9. Statistics

The GEPIA statistical analyses were performed using the ANOVA differential method, and genes with a log2 (fold change) cut-off of 2 and a *p*-value of <0.05 were considered significant. For statistical analysis in UALCAN, we used the graphs and *p*-values of the platform. We used transcripts per million (TPM) as the unit and employed their in-house PERL *t*-test. The web tool TNMplot used the Kruskal–Wallis test followed by a post hoc Dunn test. To compare the expression levels from the RNA-Seq analysis, we used log2 CPM (counts per million mapped reads) transformed values for NAT and tumoral tissue and analyzed them in GraphPad Prism 6 using an unpaired *t*-test (two-tailed, Mann–Whitney test with *p* < 0.05 considered significant). For qPCR, the difference between T and NAT was performed using the paired *t*-test (Wilcoxon test); stage comparison was analyzed using the Kruskal–Wallis test followed by the post hoc Dunn’s test.

For the correlation between gene expression level and lymph node status, we used the Mann–Whitney test on the results obtained from the qPCR analysis.

## 5. Conclusions and Perspectives

In conclusion, this study provides evidence that some TRP channels, namely, *TRPV6*, *TRPA1*, and *TRPM8*, are differentially expressed in tumoral tissues compared to normal tissues, and are linked to tumorigenesis as proven by the previous mechanistic data. Although they belong to the same protein family, and their main function is to maintain calcium homeostasis, our study sheds light on the fine modulation of these channels during tumorigenesis. The modulation comprises several proteins (especially TCAF1 and TCAF2), which participate downstream of the signaling pathways in the regulation of the invasiveness of PDAC tumors. As such, we have proven that TRPV6 interacts with TCAF1 and PTPN1; TRPA1 is related to the subfamily of TRPV channels participating in inflammation, whereas TRPM8 is connected to TCAF2 and leads towards migration pathways. Inhibition of TRPM8 and TRPA1, which are specific agonists or stimulators of *TRPV6* expression, could represent a strategy for the treatment of patients with PDAC. We envisage that the family members of the TRP family, namely, TRPV6, TRPA1, and TRPM8, and the associated TCAF1 and TCAF2 proteins, can be targets for PDAC treatment. In order to propose such important pathways for therapies, we plan to continue studying the function and pharmacology of TRP channels in cell cultures obtained from patients, and to compare the results with non-tumoral ductal pancreatic cell lines. 

## Figures and Tables

**Figure 1 ijms-23-09045-f001:**
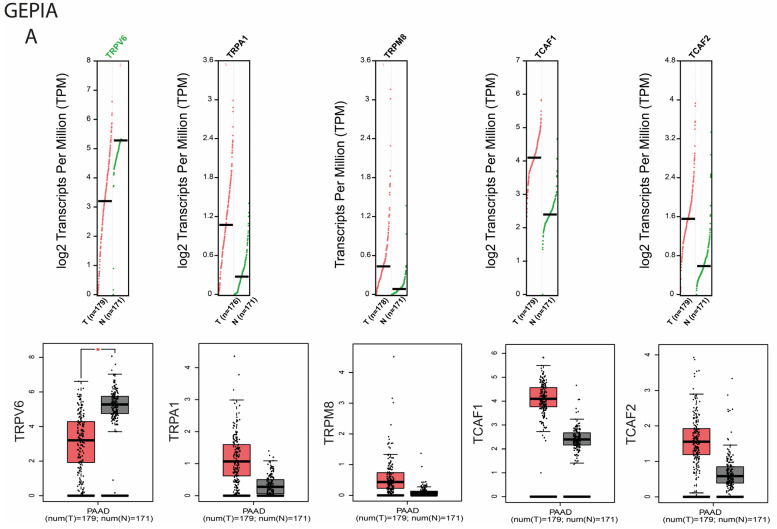
Differential mRNA expression levels of, *TRPV6, TRPA1, TRPM8, TCAF1,* and *TCAF2* in PDAC patients from public datasets. (**A**). mRNA expression levels for PDAC patients analyzed with GEPIA (red for tumor (n = 179), black for normal (n = 171)) (**B**). mRNA expression levels for PDAC patients analyzed with XENA (red for tumor, black for normal). The first and third quartiles are represented by the upper and lower horizontal lines of the box, the median by the center horizontal line, and the maximum and minimum values by the whiskers. * *p* < 0.05, **** *p* < 0.0001.

**Figure 2 ijms-23-09045-f002:**
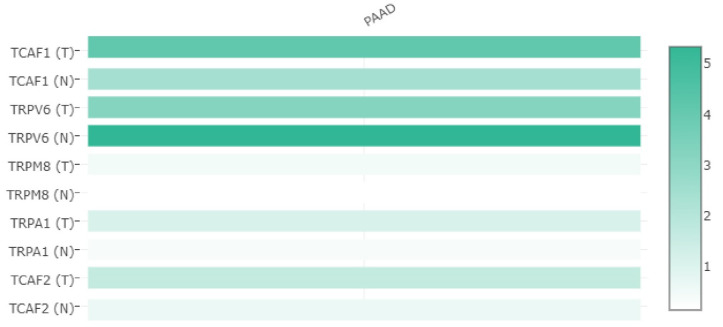
The heatmap of the selected TRP gene profiles obtained with RNA-Seq (GEPIA matrix plot). The expression profiles were obtained after normalization by log2 (TPM  +  1). The data represent a comparison of PDAC and normal pancreatic tissues from the “TCGA normal + GTEx normal” database. The color gradient spans from high expression (dark green) to low expression (light green). Log fold changes up to five-fold are illustrated, indicating differential expression with respect to the normal expression for each gene considered.

**Figure 3 ijms-23-09045-f003:**
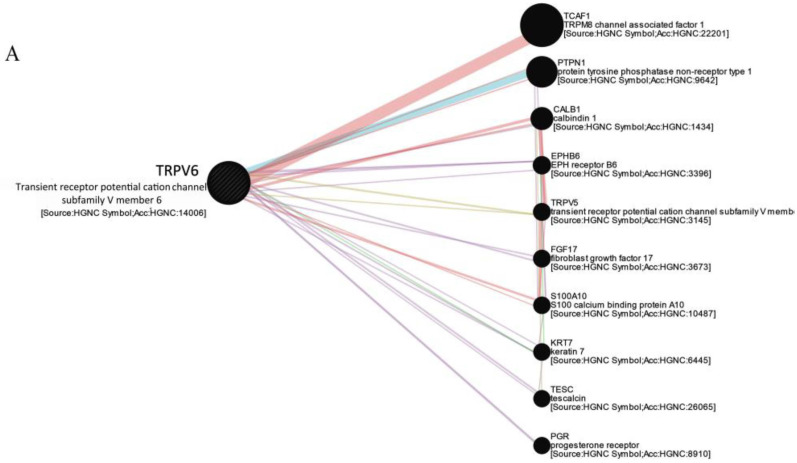
Co-expression/pathway/predicted analysis of *TRPV6* (**A**), *TRPA1* (**B**), and *TRPM8* (**C**) and 10 related genes according to human expression data in GeneMANIA (https://genemania.org/), created on 27 July 2022.

**Figure 4 ijms-23-09045-f004:**
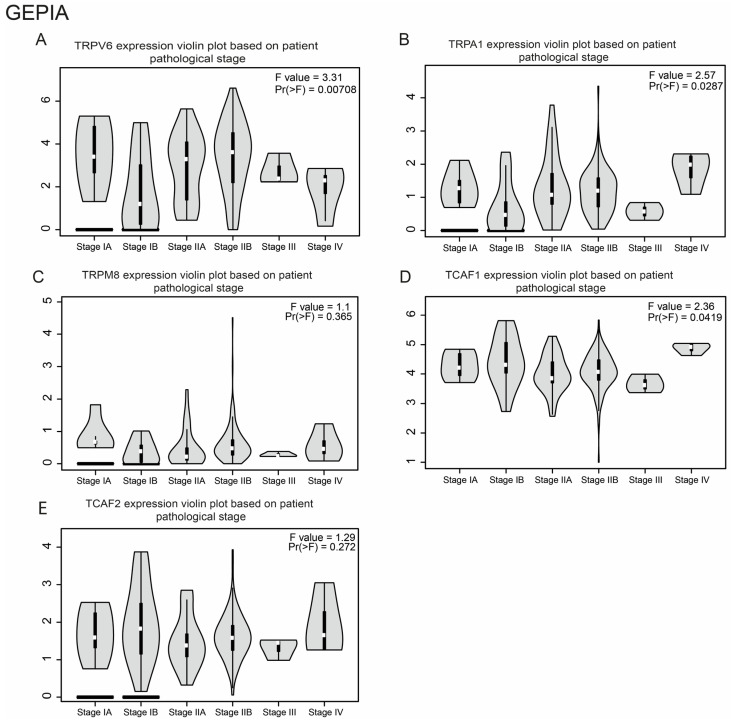
Correlation of the selected genes’ expression levels with clinicopathologic parameters. Boxplots for the correlation of *TRPV6* expression with tumoral stage ((**A**)-GEPIA, (**F**)-UALCAN, and (**K**)-TNMplot), *TRPA1* with TNM stage ((**B**)-GEPIA, (**G**)-UALCAN, and (**L**)-TNMplot), *TRPM8* ((**C)**-GEPIA, (**H**)-UALCAN, and (**M**)-TNMplot), *TCAF1* ((**D**)-GEPIA, (**I**)-UALCAN, and (**N**)-TNMplot), and *TCAF2* ((**E**)-GEPIA and (**J**)-UALCAN).

**Figure 5 ijms-23-09045-f005:**
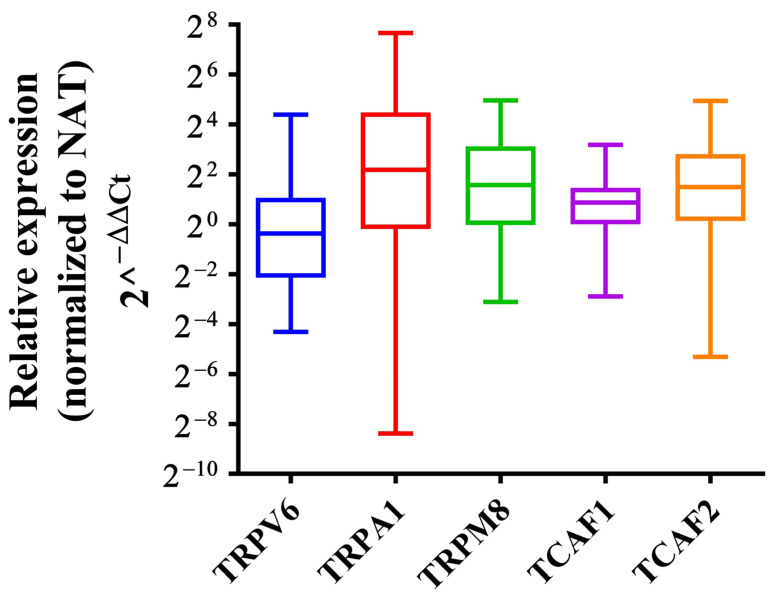
Boxplots of relative expression levels of the selected TRP genes in tissues of PDAC patients included in the validation cohort. Relative expression was calculated using the comparative Ct method (2^−ΔΔCt^) and *Hu18S* was used as the endogenous gene. Boxplots illustrate the median, lower, and upper quartiles of TRP expression in tumoral tissues normalized to NAT tissues.

**Figure 6 ijms-23-09045-f006:**
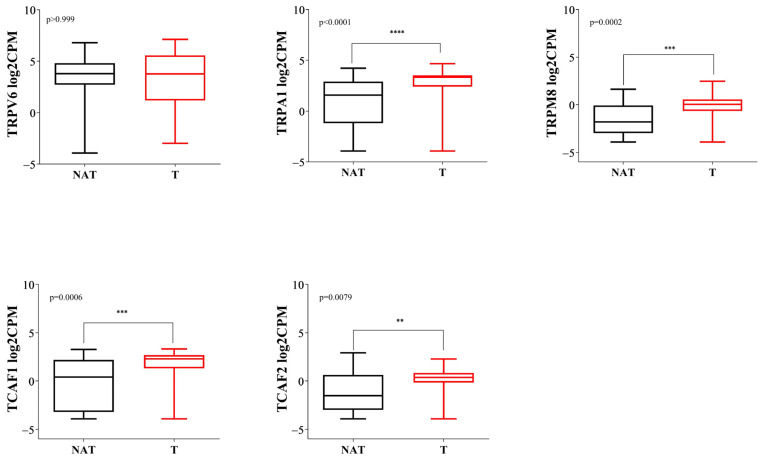
Boxplot comparing the expression levels of RNA-Seq for the selected genes in NAT vs. tumoral tissues from resected PDAC patients. The validation cohort consisted of 68 tissue samples from 36 patients (** *p* < 0.01, *** *p* < 0.001, **** *p* < 0.0001).

**Figure 7 ijms-23-09045-f007:**
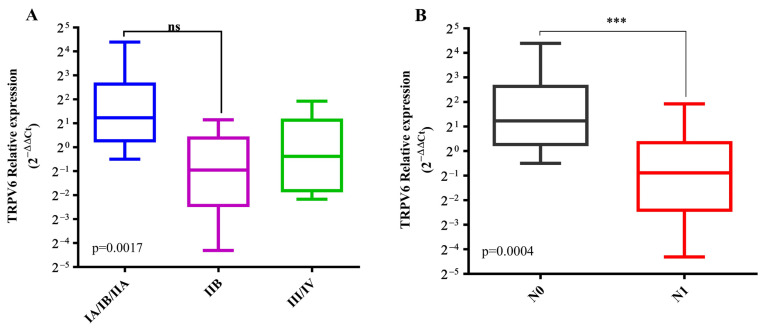
Boxplot lot comparing mRNA *TRPV6* expression in patients with PDAC according to clinicopathological variables. (**A**). Tumoral stages, (**B**). lymph node status. *** *p* < 0.001, ns indicates no statistical significance.

**Figure 8 ijms-23-09045-f008:**
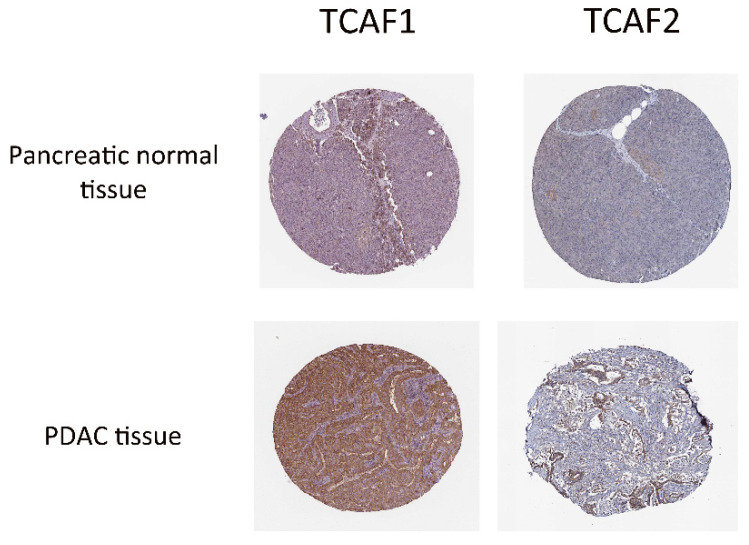
Representative immunohistochemistry images of TCAF1 and TCAF2 from PDAC tissues extracted from the HAP database.

**Table 1 ijms-23-09045-t001:** Clinicopathological characteristics of Fundeni Clinical Institute patients used for validation of the selected TRP genes by qRT-PCR and RNA-Seq.

Clinicopathological Parameters		Percentage % (n)
**Age**	≥60	61.46% (n = 67)
	<60	38.54% (n = 42)
**Gender**	Male	51.37% (n = 56)
	Female	48.63% (n = 53)
**Tumor size**	≥2 cm	73.40% (n = 80)
	<2 cm	26.60% (n = 23)
**TNM**	I	31.2% (n = 34)
	II	49.55% (n = 54)
	III	13.75% (n = 15)
	IV	5.5% (n = 6)
**Tumor differentiation**	G1	40.74% (n = 44)
	G1-G2	20.36% (n = 22)
	G2	27.78% (n = 30)
	G2-G3	11.12% (n = 12)
**CA 19-9 (U/mL)**	≥36	77.9% (n = 74)
	<36	22.1% (n = 21)
**Diabetes**	Yes	38.53% (n = 42)
	No	61.47% (n = 67)
**Vascular invasion**	Yes	42.45% (n = 45)
	No	57.55% (n = 61)
**Perineural invasion**	Yes	47.66% (n = 51)
	No	52.34% (n = 56)
**Pancreatitis**	Yes	20.44% (n = 19)
	No	79.56% (n = 74)

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
