# Peer review of "The Association between TRP Channels Expression and Clinicopathological Characteristics of Patients with Pancreatic Adenocarcinoma"

_ijms, 2022, doi:10.3390/ijms23169045_

Round 1

Reviewer 1 Report

The manuscript of Chelaru et al is an expression profile analysis of 3 TRP channels and 2 associated proteins in pancreatic adenocarcinoma. The authors use 2 different web-based tools to question public data bases and validate the differential regulation of transcripts by RNAseq and qPCR.

Even though the topic is timely and the inclusion of clinical parameters is interesting I believe further validation is needed. More precisely protein validation should be added in the paper (IHC and WB) and some functional data. For instance the calcium profile of PDAC cell lines can be studied and compared to benign or healthy ones in order to better link the profile expression data with PDAC cancer cellular and molecular aspects.

Minor comment

-change TCFA into TCAF l21

-the axis titles of figure 1, 4 & 5 charts are far too small to read

Reviewer 2 Report

This study aimed to examine to analyze the expression levels of TRPA1, TRPM8, and TRPV6 in pancreatic adenocarcinoma (PDAC) patients using data from public datasets and they also conducted transcriptomics and qPCR analyses.

This work investigates important aspects of TRP research and has clinical significance as well, although some comments and questions have arisen that need to be answered by the authors.

1.     Above all, the language of the manuscript needs to be improved. Authors should seek professional help from an expert. The name of genes and mRNA in the case of human data, need to be written with uppercase letters and italic style (e.g. TRPA1).

2.      In the Introduction, the authors provide detailed information about the PDAC and they mention the TRP channels in this regard, however, in my opinion, the role of TRP receptors in tumors in general and the rationale for this topic needs to be further elaborated together with relevant citations.

3.      Regarding Figure 1, in panel B the authors should use the same order in the case of the boxes as it was in panel A and the resolution of the images is very poor, and the labeling of the axes cannot be read. In the legend, they should specify the exact meaning of the boxes as well as the whiskers.

4.      The data in Figure 3 cannot be interpreted due to the low resolution.

5.      The authors mention the FAM115C factor’s expression in the Results section, but since FAM115C was not mentioned in the Introduction, it makes the results difficult to follow.

6.      The resolution of Figure 4 needs to be improved.

7.      In chapter 2.2.1. the authors write that they compared the gene expressions of TRPA1, TRPM8, TRPV6, TCAF, and TCAF2 to “adjacent normal tissues”. It is not clear whether they used the PDAC patient’s pancreas tissue with no macroscopic sign of a tumor, or what type of tissue? If so, how can they be sure that there were no micrometastases in this tissue?

This issue should be mentioned as a limitation of the study in the Discussion section and if they cannot prove that these types of samples were indeed intact tissues, they should avoid using the phrase “non-tumoral tissue” throughout the manuscript.

8.      In the Discussion, the authors state that “As expected, we found that TRPV6 is either co-expressed or physically related to calcium-binding protein such as calbindin 1 (CAL1) and S100 calcium-binding protein A10 (S100A10).”. My concern is that I didn't find any data on the expressions neither CAL1 nor S100A10. It is possible that this information is displayed in Figure 3 but unfortunately it has poor quality.

Reviewer 3 Report

Chelaru et al. investigated the association between the expression of TRP channels and clinicopathological characteristics in pancreatic adenocarcinoma. They assessed TRPV6, TRPA1, TRPM8, and associated regulatory factors including TCFA1 and TCFA2 using public datasets and a local validation cohort at Clinical Institute Fundeni, Romania. 

1. The authors need to use descriptive titles to their results section indicating the main finding of each result to guide readers through the paper.

2. There are many typos throughout the manuscript that require revision 

3. Higher resolution figures are needed esp. Fig. 3 

4. Several sentences are not accurate e.g., "As such, we prove that TRPV6 interacts with TCAF1 and PTPN1, TRPA1 is in relation to the subfamily of TRPV channels participating to inflammation, whereas TRPM8 is connected to TCAF2 and leads towards migration pathways". The data are correlative and are not a proof of interaction.